# Hormonal responses to brief social interactions: The role of psychosocial stress and relationship status

**Nora Nickels McLean**[1]*, **Dario Maestripieri**[2]

**1** Department of Psychological Science, Carthage College, Kenosha, Wisconsin, United States of America,
**2** Department of Comparative Human Development, University of Chicago, Chicago, Illinois, United States of America

* nmclean@carthage.edu

**Data Availability Statement:** Data cannot be shared publicly because our study protocol, which was approved by the Social and Behavioral Sciences Institutional Review Board of the

## Abstract

This study investigated the effects of psychosocial stress on hormonal responses to a social interaction with an opposite-sex individual to test the hypothesis that stress may interfere with or suppress adaptive neuroendocrine responses to courtship opportunities. Heterosexual men and women were randomly assigned prior to arrival to either a control or psychosocial stress condition (Trier Social Stress Test) and subsequently went through a social interaction test with an opposite-sex individual. Expected increases of testosterone for control participants who interacted with opposite-sex individuals were not observed, and changes in testosterone were not observed for those in the psychosocial stress condition either. However, exploratory analyses in control participants showed main and interaction effects of relationship status were significant for both cortisol and testosterone. Specifically, single individuals showed higher levels of cortisol compared to those in a relationship, and single individuals showed significantly higher concentrations of cortisol after a social interaction when compared to individuals who were in a relationship. For testosterone, only individuals in a relationship decreased in testosterone following the social interaction. This study suggests that relationship status and psychosocial stress may be important variables moderating the relationship between an ecological cue of a potential courtship opportunity and subsequent adaptive physiological responses.

## Introduction

Across many vertebrate species, testosterone regulates energy distribution and promotes investment into mating efforts in a variety of ways [1–4]. The "challenge hypothesis" [5], which has been used a conceptual framework to study the relationship between rises in androgen levels and male socio-sexual and aggressive behaviors in mating contexts, has been supported through evidence in multiple nonhuman primate and other vertebrate species [e.g., 6–8]. In humans, the importance of testosterone for male mating effort has been documented by many studies reporting on both baseline testosterone levels and testosterone responses to

University of Chicago (IRB #12-1251), does not provide for sharing de-identified participant data outside the research team, we are not able to publicly share the data. This is because data contain potentially identifying information. Data are available from the Social and Behavioral Sciences Institutional Review Board Ethics Committee of the University of Chicago (contact via 773-702-2915, or sbs-irb@uchicago.edu) for researchers who meet the criteria for access to confidential data.

**Funding:** This study was supported by the Rynerson Research Fund (NN) and Gianinno Graduate Research Fund (NN) from the University of Chicago. The funders had no role in study design, data collection and analysis, decision to publish, or preparation of the manuscript.

**Competing interests:** The authors have declared that no competing interests exist.

social stimuli [2,9–11]. The role of testosterone as a signal to coordinate behavioral investment in courtship and mate pursuit has more recently been explored in the context of the initiation of human romantic relationships [9]. Specifically, numerous studies have examined the reactive testosterone responses of heterosexual male individuals to naturalistic social interactions with female individuals [12–20].

Roney and colleagues [17] were some of the first to test for hormonal reactions of human men to brief social encounters with opposite-sex individuals, considered to be potential mating partners. They found that salivary testosterone of young men increased significantly over baseline levels after engaging in a short conversation with a young woman (this effect was only evident, or was most pronounced, in single men than in men in stable romantic relationships; see below) but did not increase significantly over baseline levels after engaging in short conversation with another young man [17,18]. Consistent with the notion that dominance may also play a role in fluctuations of testosterone during brief encounters with a potential mate, van der Meij et al. [20] found that salivary testosterone increases after brief interactions with women were most evident in men with aggressive dominant personalities, men who were not involved in committed, romantic relationships, and men who had been sexually inactive for over a month. Therefore, individual differences in personality, relationship status, and sexual motivation may affect testosterone reactivity that occur in response to courtship opportunities.

Lopez et al. [21] replicated in women the findings of Roney and colleagues [17,18]. Female participants showed significant reactive increases in testosterone after viewing a movie clip of a courtship situation with an attractive man and a young woman [21]. It is possible that female-female competition for potential courtship opportunities may induce a similar testosterone increase in women to aid in self-confidence, to preserve dominance and status, and to emphasize motivation to affiliate with potential mates. Although most previous studies of testosterone and competition have been conducted with men, there is some evidence that similar effects can be observed in women as well [22,23]. Researchers are currently calling for further research on the endocrinology of competition and courtship in women and are emphasizing the importance of including female participants in study designs [24,25]. The purpose of the current study includes observing potential short-term androgen increases in both men and women.

All social evaluative interactions involve stressful components, and courtship in particular may cause significant apprehension during courtship interactions [26–28]. We know that human courtship behavior did not evolve in a stress-free environment, and that any kind of social evaluation places pressures upon and threatens one's own self perceptions, through dampening self-motivation, self-confidence, or perception of social status. Further, the "dual hormone hypothesis" postulates that testosterone's association with status-relevant and sociosexual behavior depends on cortisol, a hormone released during physical and psychological stress [29–32]. Therefore, it is important that we consider how stress may affect courtship behavior in either adaptive or maladaptive ways. The purpose of the current study also includes observing the effects of stress on androgen reactivity to potential mating scenarios.

Cortisol is a glucocorticoid hormone released by the hypothalamic-pituitary-adrenal (HPA) axis in response to stress as part of its broader role as a physiological regulator of blood pressure, metabolism, and energy mobilization [33,34]. Much research places the focus of cortisol increase as a coping mechanism to aversive physical or psychological conditions and as an adaptive function to divert energy and suppress unessential processes [34]. Importantly, cortisol responses are also elicited when one is faced with a psychological threat to one's self esteem and when psychological well-being is threatened [35]. We could potentially expect courtship apprehension to trigger a cortisol increase in addition to a testosterone increase that may prove to alter courtship behaviors.

There is already some indirect evidence for the role of the HPA axis in human courtship, as displayed by cortisol increases that could reflect anxiety or apprehension before or during opportunities for courtship [18,19,36,37]. In multiple experiments, Roney et al. [18] found that changes in cortisol from baseline were significantly greater among male participants who interacted with female individuals relative to male participants in control conditions. Lower baseline cortisol concentrations have also been shown to predict larger testosterone responses to interactions with potential female mates [15], consistent with the dual hormone hypothesis. Similarly, van der Meij and colleagues [36] found that cortisol levels of men increased when they interacted with a female individual whom they reported as attractive. Beyond a one-on-one social interaction, cortisol has been found to be sensitive to the sex composition of the environment [38] and to anticipatory social exchanges where participants perceive potential interactions as occurring with a desirable dating partner [39].

It remains unclear overall how stress during courtship and subsequent increases in cortisol may interfere with or trigger other physiological and psychological responses to courtship [see 40, for different effects of psychosocial stress on cooperative behavior in male and female participants]. For example, an increase in cortisol may subdue or promote appropriate courtship behaviors based on the profile of testosterone reactivity, as predicted by the dual hormone hypothesis. If testosterone shows acute increases in response to a courtship opportunity, despite a simultaneous increase in cortisol, courtship behaviors may not be hindered in a maladaptive way. However, if a cortisol increase is accompanied by a testosterone decrease, stress may prove to interfere with subsequent motivation, self-confidence, and affiliative behaviors.

It is currently uncertain whether cortisol responses are generally associated with specific courtship behavior patterns, or whether the association is an effect that is strictly mediated by motivation, cognition, or individual differences in personality or relationship status. Further, anxiety itself may drive the physiological responses of testosterone, in addition to its effects on cortisol. The effects of stress on testosterone may depend specifically on the type of social stress induced and may mediate the relationship between individual differences and behavior. For example, Crowley et al. [41] found that testosterone mediated the relationship between uncertainty and the amount of disclosure between romantic partners.

The goal of the present study was to investigate testosterone and cortisol responses to social interactions with opposite-sex individuals in both men and women, with or without a prior exposure to psychosocial stress. Primary a priori hypotheses involved the effects of social interactions with opposite-sex individuals on testosterone in men and women, and we explored these hypotheses using two different psychosocial tasks. The first was a non-stressful social interaction task with an opposite-sex individual, which all participants experienced. The second was a psychosocial stress task that half of the participants experienced prior to their non-stressful social interaction with an opposite-sex individual. First, we predicted that we would replicate findings of increases in testosterone for both male and female participants during brief interactions with opposite-sex individuals. This increase would be expected specifically for individuals who have *not* gone through a psychosocial stress manipulation, replicating past studies that have found testosterone increases for men and exploring the possibility that this testosterone increase may be seen in both men and women. Second, we predicted that a psychosocial stress manipulation could alter a testosterone response in either of the following ways: 1) if cortisol increases dramatically due to psychosocial stress, testosterone increases could be dampened, and 2) if cortisol reactivity to stress is low, this may be associated with higher increases in testosterone. In addition to these main research questions, we planned exploratory analyses on whether testosterone increases would be most apparent in individuals whose motivations are in line with potential courtship behaviors (e.g., single individuals), due to lack of direct evidence for such findings in humans. Cortisol responses to social interactions

as a function of relationship status are also considered exploratory, given mixed findings in nonhuman literature and the lack of direct evidence in studies using both men and women.

## Method

### Participants

156 participants (62 men, age: $M = 22.56$, $SD = 4.37$; 94 women, age: $M = 21.89$, $SD = 3.37$) were recruited from the University of Chicago campus and surrounding area through fliers, UChicago Marketplace, and a human subject recruitment website. Participants signed an electronic copy of the consent form prior to arrival and were paid $20 after completion of the study.

### Experimental procedure

In this study, the eligibility criteria for participant recruitment were age (between 18 and 35 years) and heterosexual orientation. All experimental procedures were approved by the Social Science Institutional Review Board at the University of Chicago. All experimental procedures took place between 11:30 AM and 5:30 PM. Participants always interacted with an experimenter, or "greeter", of the same sex throughout the entire experimental session. Upon arrival to the research building, participants were taken by a "greeter" to the testing room, where they completed questionnaires for 20 minutes. An initial demographic survey asked information about participants' age, ethnicity, sexual orientation, socioeconomic status, marital or relationship status (single or in a relationship), etc. At the end of this period, they provided a baseline saliva sample. They then either took part in the Trier Social Stress Test or sat in a room doing nothing for a similar period of time as a control condition. Another saliva sample was collected after the TSST or the control condition (post-treatment), approximately fifteen minutes after the start of the TSST or control condition. Approximately ten to fifteen minutes after the TSST or control condition had ended, participants went through a brief social interaction task with a research assistant of the opposite sex. A final saliva sample (post-social interaction) was collected after the social interaction task. Upon completion of all procedures, participants were fully debriefed and given compensation.

### Trier social stress test

The Trier Social Stress Test (TSST) [42] is a broadly used, standardized task that is used to study hormonal responses to mild psychosocial stress in a laboratory setting. In the current study, the experimenter explained to each participant that he or she would be giving a 5-minute presentation about himself or herself for a mock job interview. Each presentation took place in front of a "selection committee" composed of two unfamiliar judges trained to maintain neutral facial expressions and provide no positive feedback to the participant. Each participant was informed that he or she must keep speaking for 5 minutes and that the presentation would be video-recorded for subsequent analyses of content and non-verbal behavior. If the participant ever stopped speaking before the 5 minutes were up, the judges waited in silence for the participant to resume or otherwise prompted him or her to continue. If the participant again stopped speaking, one of the judges asked one of several standardized questions (e.g., "What do you think about teamwork?"). Upon completing the 5-min speech, the judges asked each participant to perform a difficult arithmetic calculation (i.e., serially subtracting the number 17 from 2,023) out loud for another 5 minutes or until he or she reached zero. Anytime the participant made a mistake, he or she was notified and asked to restart from the beginning. After this task, the judges thanked the participant and left the room.

Although the "greeter" who interacted with the participant was always the same sex as the participant, the TSST judges were either of the same or the opposite sex as the participant, in alternation. Therefore, female participants were assigned to interact with either two same-sex TSST judges (a female "talking judge" and a female "timing judge"), or two opposite-sex TSST judges (a male "talking judge" and a male "timing judge"). Likewise, male participants were assigned to interact with either two same-sex TSST judges (a male "talking judge" and a male "timing judge"), or two opposite-sex judges (a female "talking judge" and a female "timing judge"). The semi-random assignment of the judges was based on availability of the research assistants but was also counter-balanced as much as possible. Over the fourteen months of data collection, judges and greeters were assigned based on availability from a team of eighteen different undergraduate research assistants (age range 18–27; 46.7% female).

Participants who were assigned to the control condition and who did not participate in the TSST simply sat by themselves for 10 minutes until their original experimental "greeter" returned to let them know they could continue moving forward in the study.

## Social interaction task

Following the control/TSST condition, all participants partook in a social interaction task, where they interacted with an opposite-sex research assistant whom they had not encountered yet at that point in the session. The social interaction task that was used in this experiment was adapted from brief social interaction tasks that have been used in several research studies in which a social inter-action involving a research assistant posed as either another participant or experimenter led to physiological and behavioral changes [17,18]. In our study, the experimental "greeter" let the participant know that they needed approximately five or ten minutes to pass before moving on to the next part of the study, and that the participant was free to relax until the experimenter returned. Several minutes after the departure of the "greeter", an opposite-sex researcher entered the room and introduced himself or herself as a research assistant there to collect data off of a digital video camera (earlier in the session, this video camera was used to collect a digital photograph of every participant, as well as used to record the TSST session for participants assigned to the TSST condition). Chairs were arranged in the room such that the participants always sat directly across from the researcher with a small conference table positioned between them. Research assistants then attempted to engage in natural, friendly conversation, while simultaneously uploading data from the digital video camera onto a computer or hard drive. The research assistants were free to use whatever means of engaging in conversation seemed natural to them. Scripts or specific prompts were not used to avoid interactions seeming excessively artificial. Conversations lasted seven minutes, at which point the experimenter re-entered the room and interrupted the research assistant and participant to seemingly complete the rest of the study protocol. As with greeters and judges, conversation partners were assigned based on availability from a team of eighteen different under-graduate research assistants (age range 18–27; 46.7% female). Participants on average rated the opposite-sex research assistant slightly above (i.e., slightly above a rating of 4) or slightly below (i.e., slightly below a rating of 4) the midpoints of the seven-point scales that assessed perception of physical attractiveness ($M = 4.33$, $SD = 1.60$), desirability as a short-term romantic partner ($M = 3.32$, $SD = 1.78$), and desirability as a long-term romantic partner ($M = 3.34$, $SD = 1.70$). Such ratings suggest that the participants did not have particularly strong interest in the opposite-sex research assistant but did rate individuals as above average in terms of attractiveness.

## Saliva sample collection and hormonal assays

All saliva samples were collected between 12:00 PM and 5:00 PM, as previous studies have shown that afternoon cortisol and testosterone levels, although lower than morning levels, are

more stable and therefore better suited for studies of social endocrinology [35,43–46]. Strong circadian variation is observed for both cortisol and testosterone, and by conducting testing at just one time of day (i.e., in the afternoon), researchers can avoid some of the variance determined through circadian fluctuation [44]. In terms of sampling, intervals of 15 minutes are commonly used due to the time it takes for steroid hormones to transfer from blood to saliva after negative feedback loop release [44]. Saliva was collected by passive drool into plastic tubes. Saliva samples were stored in a refrigerator at -20˚F. Samples were assayed for testosterone and cortisol concentrations using ELISA kits purchased from Salimetrics. For both cortisol and testosterone assays, samples competed with hormones conjugated to horseradish peroxidase, and assay sensitivity was <0.007 ug/dL and 1 pg/mL, respectively. Saliva sample concentrations were calculated based on kit standards using a 4-parameter nonlinear regression curve fit. For cortisol, the average inter-assay CV based on concentration was 4.85% (3.54% for high controls; 6.17% for low controls), and the intra-assay CV based on concentration was 7.15%. For testosterone, the inter-assay CV based on concentration was 6.63% (4.86% for high controls; 8.40% for low controls), and the intra-assay CV based on concentration was 5.38%. CVs were calculated from the concentrations rather than raw optical densities. Specifically, the inter-assay CV was calculated using the mean values for the quality high and low controls on each plate, and the intra-assay CV was an average value calculated from the individual CVs from all the plate duplicates. The cutoff for reanalysis used was 10%; any duplicate above this cutoff were removed and reanalyzed. In the overall sample, baseline cortisol and testosterone levels were in line with assay protocol salivary example PM range norms.

## Data analyses

Four individuals did not complete the full protocol due to researcher error, participant withdrawal from the study, or computer error, and therefore were not included in data analyses. Two more individuals were excluded from all hormonal analyses due to having baseline or delta hormonal concentrations over three standard deviations away from the mean. One participant was excluded from all hormonal analyses due to saliva samples being heavily contaminated with blood. After excluding these seven individuals, data were analyzed for a total of 149 individuals, of whom 76 (46 women, 30 men) underwent the TSST stress manipulation and 73 (47 women, 26 men) were in the control condition. Of all participants, 81 indicated their relationship status as single, and 68 indicated their relationship status as in a relationship at the time of the study. Of all TSST participants, 40 were single and 36 were in a relationship; of all control participants; 41 were single and 32 were in a relationship.

All statistical analyses were carried out with jamovi software (Version 2.2.5.0) [47]. When hormonal data were not normally distributed, they were log transformed. When sphericity assumptions were violated, Greenhouse-Geisser corrected p-values were reported. Alpha was set at 0.05 and adjusted for multiple comparisons where necessary.

## Results

### Manipulation check

To confirm the effects of the TSST on participants' stress levels, the effect of the TSST manipulation on physiological stress (cortisol) was tested with a 2 (treatment group: control vs. TSST) x 3 (timepoint: baseline, post-treatment condition, post-social interaction) mixed model, with timepoint as a repeated measure and cortisol concentration as the outcome variable. A Greenhouse-Geisser correction was applied since the assumption of sphericity was not met. An expected interaction effect between treatment group and timepoint was significant [$F_{(1.71,251.13)} = 14.58$, $p < 0.001$, $\eta^2_P = 0.09$ (a moderate effect); Fig 1]. Specifically, individuals

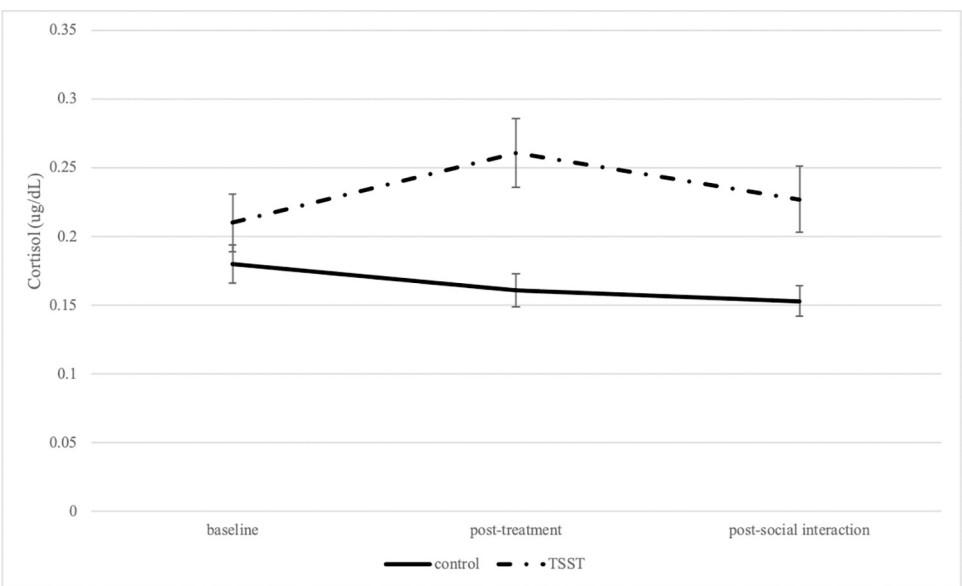

**Fig 1. Changes in cortisol concentrations between control and TSST participants.** Salivary cortisol concentrations (measured at baseline, post-treatment, and post-social interaction) depicted by experimental condition. Values are mean ± SEM.

who underwent the TSST had higher cortisol concentrations post-treatment ($t$ = -4.48, $p<0.001$) and post-social interaction ($t$ = -3.73, $p = 0.004$).

## Testosterone responses to social interaction

The effects of the social interaction and TSST manipulation on testosterone was tested with a 2 (timepoint: pre-social interaction and post-social interaction) x 2 (treatment group: control vs. TSST) mixed model, with timepoint as a repeated measure and testosterone concentration as the outcome variable. Unlike what we hypothesized, there was no significant interaction effect between timepoint and treatment group [$F(1,147)$ = 0.08, $p = 0.77$; Fig 2]. Specifically, there were no changes in testosterone concentrations for participants who were not exposed to psychosocial stress before a social interaction with an opposite-sex research assistant, and there were no changes in testosterone concentrations for participants who were exposed to psychosocial stress before a social interaction with an opposite-sex research assistant. No main effects for timepoint [$F(1,147)$ = 0.58, $p = 0.45$] or treatment [$F(1,147)$ = 0.003, $p = 0.95$] were found.

## Exploratory analyses of relationship status

To explore how relationship status relates to hormone reactivity, subgroup analyses were performed separately in control and TSST participants. First, the effect of relationship status on testosterone responses in control participants was tested with a 2 (relationship status: single vs. in a relationship) x 2 (timepoint: pre-social interaction and post-social interaction) mixed model, with timepoint as a repeated measure and testosterone concentration as the outcome variable. No main effects of relationship status [$F(1,71)$ = 1.25, $p = 0.27$] or timepoint [$F(1,71)$ = 1.39, $p = 0.25$] were found. An interaction effect between relationship status and timepoint was found [$F(1,71)$ = 11.45, $p = 0.001$, $\eta^2_P$ = 0.14 (a large effect); Fig 3], such that participants in a relationship had lower concentrations of testosterone after the social interaction when compared to prior to the social interaction ($t$ = 3.04, $p = 0.02$). However, the hypothesized testosterone increase in single individuals after exposure to social interaction was not observed.

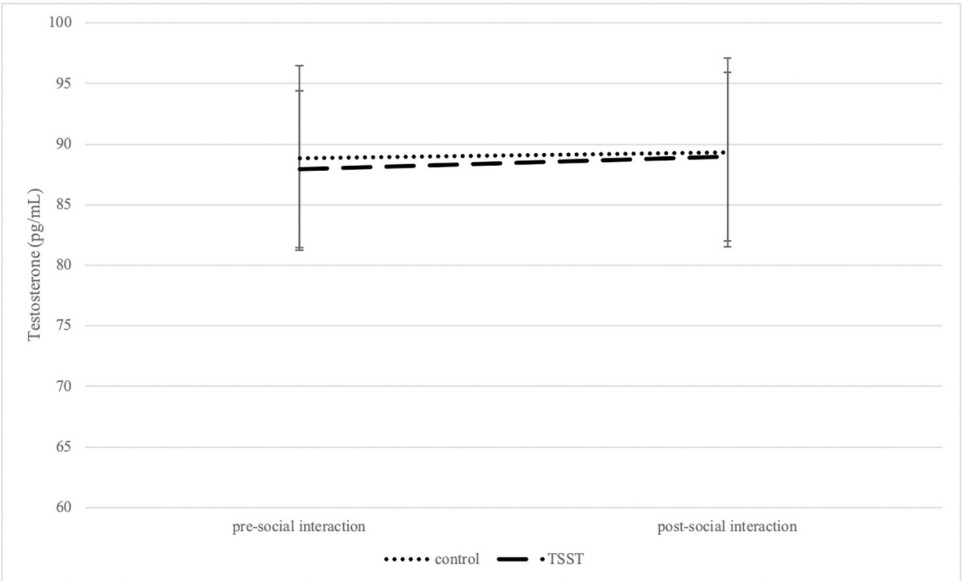

**Fig 2. Changes in testosterone concentrations following social interaction as a function of treatment.** Salivary testosterone concentrations (measured at pre-social interaction and post-social interaction) depicted by experimental condition. Values are mean ± SEM.

The effect of relationship status on cortisol responses in control participants was tested with a 2 (relationship status: single vs. in a relationship) x 2 (timepoint: pre-social interaction and post-social interaction) mixed model, with saliva sample as a repeated measure and cortisol concentration as the outcome variable. An interaction effect between relationship status and timepoint was found [$F(1,71) = 10.69$, $p = 0.001$, $\eta^2_P = 0.13$ (a medium effect); Fig 4]. Specifically, participants in a relationship had lower concentrations of cortisol after the social

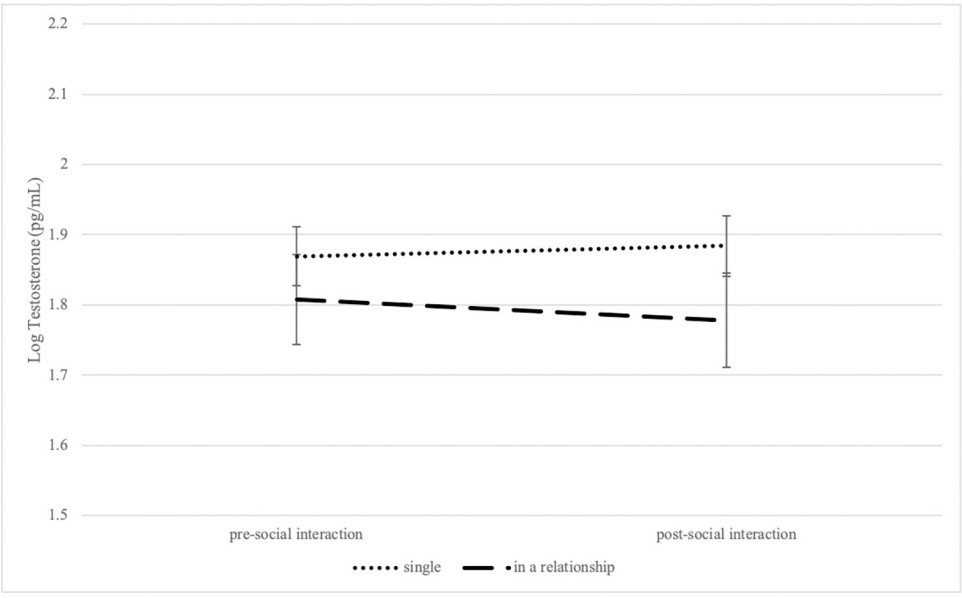

**Fig 3. Changes in testosterone concentrations following social interaction as a function of relationship status in control participants.** Salivary testosterone concentrations (measured at pre-social interaction and post-social interaction) depicted by relationship status. Values are mean ± SEM.

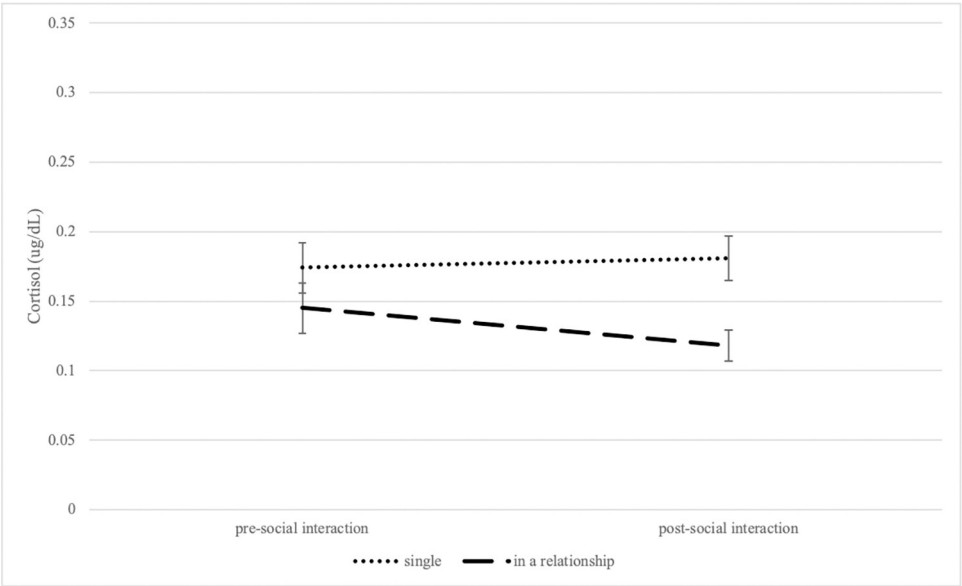

**Fig 4. Changes in cortisol concentrations following social interaction as a function of relationship status in control participants.** Salivary testosterone concentrations (measured at pre-social interaction and post-social interaction) depicted by relationship status. Values are mean ± SEM.

interaction when compared to prior to the social interaction ($t = 3.13$, $p = 0.02$), and participants in a relationship had lower concentrations of cortisol after the social interaction when compared to single participants after the social interaction ($t = 3.33$, $p = 0.008$). A main effect of relationship status was also found [$F(1,71) = 5.67$, $p = 0.02$, $\eta^2_P = 0.07$ (a medium effect)], such that control participants in a relationship had lower levels of cortisol overall when compared to single participants ($t = 2.38$, $p = 0.02$). No main effect of timepoint was found [$F(1,71) = 2.02$, $p = 0.16$].

Exploratory analyses of relationship status were also conducted for participants who went through the TSST stress condition. The effect of relationship status on testosterone responses in TSST participants was tested with a 2 (relationship status: single vs. in a relationship) x 2 (timepoint: pre-social interaction and post-social interaction) mixed model, with timepoint as a repeated measure and testosterone concentration as the outcome variable. No significant main effects of timepoint [$F(1,74) = 0.063$, $p = 0.803$] or relationship status [$F(1,74) = 0.00$, $p = 0.99$] were found. The interaction effect between timepoint and relationship status was also not significant [$F(1,74) = 1.54$, $p = 0.22$]. For cortisol, an expected significant main effect of timepoint was found in TSST participants [$F(1,74) = 13.58$, $p<0.001$, $\eta^2_P = 0.16$ (a large effect)], such that participants post-social interaction had lower cortisol concentrations when compared to participants pre-social interaction ($t = 3.69$, $p<0.001$); this decrease was expected, as it was likely that participants' cortisol concentrations were decreasing after the post-TSST increase. Similar to subgroup analyses with control participants, a significant main effect of relationship status was observed [$F(1,74) = 5.17$, $p = 0.03$, $\eta^2_P = 0.07$ (a medium effect)], such that TSST participants in a relationship had lower levels of cortisol overall when compared to single participants ($t = 2.27$, $p = 0.03$).

## Discussion

The present study investigated hormonal responses to brief social interactions following social evaluative stressors. We hypothesized that individuals would show increases in testosterone

following brief social interactions and that exposure to prior social evaluative threat and relationship status would moderate these increases. However, among all control male and female participants, we did not find significant increases in testosterone after taking part in a brief conversation with a researcher of the opposite sex. Further, the psychosocial stress treatment did not moderate or cause changes in testosterone following the social interaction. Exploratory subgroup analyses did show a difference in testosterone at differing time points for individuals who were in a relationship for control participants. Specifically, individuals in a relationship had significantly lower testosterone concentrations after the social interaction compared to testosterone concentrations prior to the interaction. These results do not replicate those of previous studies showing that in single men, verbal interactions with women (of less than five or ten minutes) can causally induce increases in testosterone [17,20]. However, the results may be consistent with past findings that have shown singles to have higher testosterone concentrations [48] and with findings that testosterone concentrations are higher after a social interaction for those who are single when compared to those who are in a relationship [17,20]. Overall, these results only partly support the challenge hypothesis. General increases in testosterone were not observed in men and women during psychosocial encounters, yet it is possible that the significant findings via exploratory analyses focusing on relationship status provide evidence that motivational differences could lead to differences in testosterone reactivity that correspond with appropriate socio-sexual behavior.

Similar to testosterone, exploratory subgroup analyses showed differences in cortisol between single individuals and individuals in a relationship for control participants. Specifically, singles showed significantly higher cortisol concentrations when compared to individuals who were in a relationship overall, and single individuals had higher concentrations of cortisol after the social interaction when compared to individuals in a relationship. These results are partly consistent with those of previous studies in suggesting that cortisol increases could reflect anxiety or apprehension before or during opportunities for courtship [18,19,36]. For example, Roney et al. [18] found that changes in cortisol from baseline were significantly greater among male participants who interacted with women relative to men in control conditions. Specifically, Roney et al. [18] and van der Meij et al. [36] found that singles showed greater hormonal reactivity when compared to individuals in a relationship. These results are also consistent with past findings that have shown singles to have higher cortisol concentrations [49].

Further, these results are consistent with an article by Zilioli and Bird [50], who after reviewing the literature on testosterone reactivity (primarily in human men), concluded that both motivational factors and situational factors have the ability to influence the relationship between evolutionarily salient social contexts, such as an interaction with a potential mate, and physiological reactivity to these contexts [50]. Differences in relationship status could, at least in part, reflect differences in socio-sexual motivation. A situational factor, which the authors define as an external factor outside of the control of the individual, could influence the relationship between ecological cue and physiological response [50]. In our study, psychosocial stress may have interfered with the physiological response to social stimuli in control participants, which explains why the significant changes in testosterone in individuals in a relationship were no longer observed in TSST participants. This possible interference may also be evidence of the dual hormone hypothesis; an increase in cortisol may have subdued the testosterone reactivity observed in control participants in a relationship. However, our results did not support the expected evidence of the dual hormone hypothesis whereby changes in cortisol altered testosterone reactivity meant to promote appropriate courtship behavior, as expected increases in testosterone were not observed in control participants. Future research on the relation between stress and courtship is needed, especially in light of recent studies reporting anxiety-related reactions during first encounters with individual perceived as attractive [51].

Our interpretation of the results is tentative, and we acknowledge that our study has some methodological limitations. First, previous laboratory studies of human courtship have used highly attractive research assistants as social interaction partners, who engaged in friendlier, more deliberately flirtatious conversations [19], and some field studies often increase validity by measuring hormones before, during, and after dating events [36]. As this study chose to keep the social interaction more neutral, this may have resulted in fewer participants interpreting the encounter as a potential courtship opportunity, and we did not ask participants if they were given the impression to flirt from the research assistant. This difference may in part account for the discrepancy between these results and those of previous studies. Further, we did not use a control condition in which participants either engaged in conversation with a same-sex social interaction partner or simply did not participate in conversation at all. Second, although it remains crucially important for future research to focus on context-dependent acute hormonal changes in women, recent work has pointed out that it is challenging to measure variability in testosterone concentrations in women using standard enzyme immunoassays (EIAs), which may tend to inflate estimates of lower concentration estimates in women [52]. Despite the ease and cost-effectiveness of EIAs, we encourage further research using liquid chromatography mass spectrometry to measure hormone concentrations, as this methodology may be free from some of the limitations involved with EIAs [52]. Third, budgetary restrictions limited the number of hormone samples taken throughout the experimental procedure. Only one baseline hormonal assessment and one test assessment after the TSST / control were measured, and allowing time for hormone levels to return back to baseline after the experimental treatment could have simplified the interpretation of our results.

Overall, this study highlights the importance of individual differences in relationship status and situational variables as factors in moderating hormonal reactivity to a potential courtship interaction. It appears that psychosocial stress, while not affecting testosterone reactivity during a social interaction directly, may play a role in altering how men and women react to a social interaction interpreted as a potential courtship opportunity. While the exact mechanism remains unclear, it is possible that psychosocial stress may suppress the adaptive physiological response (increase in testosterone) in different ways dependent on relationship status for individuals who are engaged in a courtship. Future studies should continue to focus on the potential moderating factors, whether biological, sociological, or psychological in nature, as there is growing evidence that these factors have the potential to influence the relationship between evolutionarily salient social contexts (e.g., interaction with mates) and adaptive neuroendocrine responses.

## Supporting information

**S1 File. ZIP file containing data analysis script (.omt files) for three sets of analyses.** (ZIP)

## Acknowledgments

We thank Jim Roney for his helpful feedback on this manuscript. We thank Mint Poonpatanapricha, Coltan Scrivner Cameron Miller, Yanitza Roman, Kay Yang, Melanie Sykes, Alex Portee, Ian McCann, Jonah Lowenstein, Joseph Wiltzer, John Luo, Justin Pan, Emil Qiu, Antonia Theodosakis, and Kat Pervova for their assistance with data acquisition.

## Author Contributions

**Conceptualization:** Nora Nickels McLean, Dario Maestripieri.

**Data curation:** Nora Nickels McLean, Dario Maestripieri.

**Formal analysis:** Nora Nickels McLean.

**Funding acquisition:** Nora Nickels McLean.

**Methodology:** Nora Nickels McLean, Dario Maestripieri.

**Resources:** Dario Maestripieri.

**Supervision:** Dario Maestripieri.

**Writing – original draft:** Nora Nickels McLean.

**Writing – review & editing:** Dario Maestripieri.

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
