## [Decision Letter · Decision Letter 0]

16 Mar 2022

PONE-D-21-20363Hormonal responses to brief social interactions: The role of psychosocial stress and relationship statusPLOS ONE

Dear Dr. Nickels,

thank you for submitting your manuscript to PLOS ONE and your patience while awating the completion of reviewing process.  I am happy to confirm the arrival of Reviewers' reports on your submission.  After careful consideration, we feel that your manuscript has merit but has to be revised to fully meet PLOS ONE’s publication criteria. Therefore, we invite you to submit a revised version of the manuscript that addresses the points raised during the review process. Please make sure to carefully consider and respond to Reviewers' concerns in the point-by-point manner, paying particular attention to (1) the technical aspects of your work, (2) statistical issues, (3) justification of your conclusions by the data, and (4) data availability as mentioned in the comments below.  Please submit your revised manuscript within six months from this date as after that point any revision has to be considered a new submission. If you will need more time than this to complete your revisions, please reply to this message or contact the journal office at plosone@plos.org. Please include the following items when submitting your revised manuscript:A rebuttal letter that responds to each point raised by the academic editor and reviewer(s). You should upload this letter as a separate file labeled 'Response to Reviewers'.A marked-up copy of your manuscript that highlights changes made to the original version. You should upload this as a separate file labeled 'Revised Manuscript with Track Changes'.An unmarked version of your revised paper without tracked changes. You should upload this as a separate file labeled 'Manuscript'.

We look forward to receiving your revised manuscript.  Thank you for choosing PLOS ONE as outlet for reporting your reseach.

Kind regards,

Sasha

Alexander N. 'Sasha' Sokolov, Ph.D.

Academic Editor

PLOS ONE

Journal Requirements:

2. Please change "female” or "male" to "woman” or "man" as appropriate, when used as a noun (see for instance https://apastyle.apa.org/style-grammar-guidelines/bias-free-language/gender).

Reviewers' comments:

Reviewer's Responses to Questions

**Comments to the Author**

1. Is the manuscript technically sound, and do the data support the conclusions?

Reviewer #1: Partly

Reviewer #2: Yes

2. Has the statistical analysis been performed appropriately and rigorously? 

Reviewer #1: No

Reviewer #2: No

3. Have the authors made all data underlying the findings in their manuscript fully available?

Reviewer #1: Yes

Reviewer #2: No

4. Is the manuscript presented in an intelligible fashion and written in standard English?

Reviewer #1: Yes

Reviewer #2: Yes

5. Review Comments to the Author

Reviewer #1: In this study, the authors investigate the effect of psychosocial stressors on hormonal reactivity to brief encounter with opposite sex individuals in heterosexual humans. The study appear relatively well designed and well suited to test the general question. However, it appears a bit unclear when reading the paper what the question really is. The authors mix up hypothesis and predictions and provide a list of examples of previous studies rather than clear testable hypothesis. Furthermore the statistical analyses are outdated (I strongly recommend the use of Generalized Linear Mixed Models) and some key variable appear uncontrolled for (see below).

Introduction

The introduction is relatively lengthy, not necessarily because it is too long but because it presents a suit of detailed findings from other studies with no clear direction of where the authors go. The main theories regarding hormonal responsiveness to social interactions are not named and the main theories published are not cited. The authors should revise the introduction to expand on theory and merge the exposition of previous findings into less lengthy phrasing.

In the introduction the interplay between Testosterone and cortisol in their hormonal regulation and the negative feedback effects that cortisol has on testosterone should be presented and discussed and incorporated in the hypothesis.

Line 47: please nominatively introduce the challenge hypothesis (Wingfield et al. 1990) and expand citation to other studies which directly cited it in non-human primates or other animals (e.g. Muller & Wrangham 2004; Hirschenhauser & Oliveira 2006; Girard-Buttoz et al. 2015). One key aspect of this hypothesis is differences in testosterone reactivity to social interactions depending on paternity status. It appears therefore crucial to control for this parameter in the analysis beyond simply assessing if individuals are single or not.

Line 75: Here the authors point out the important effect of anticipation. However this parameter was not tested in their experiment since the participants did not know that they would be placed in a (albeit neutral) courtship opportunity situation. Such anticipatory principle is therefore not relevant to the current study.

Line 93: Define ‘confederates’.

Line 121: present what cortisol is and highlight that it regulates the hormonal stress response in mammals including humans. In general, beyond providing a list of what testosterone and cortisol do and have been found to do it would be useful to provide clear cut presentation of what they regulate (e.g. muscle mass, aggressiveness for testosterone and energy mobilization for cortisol) and what they are expected to be sensitive too and why.

Line 123: What is presented in the following lines are not hypothesis but prediction and testable patterns… please formulate hypothesis first.

Methods:

The methods are overall clear and well detailed. As previously, mention the authors should use GLMM rather than anova to clarify methodology and include more control variables. The authors should also run first a full model with all the interactions considered and compared it to a null model using an ANOVA and then reduce this full model removing non-significant interactions to obtain a final reduced model where the interactions and single effects are interpretable. In fact, single effects in interactions cannot be interpreted on their own. As previously mentioned, the authors should control for some important factors such as the paternity status of the men and the maternity status of the woman as well as whether they failed or passed the mathematical test which may have played a role in their hormonal stress response (one can imagine that failing is more stressful and that individual who failed may have more struggle solving mathematical problems hence be exposed to a higher stress).

Line 135: Write men and woman rather than male and female, it should be clear right from the start that you test humans.

Line 147: Clarify what is SES

Line 217: Hormonal level is too general please indicate which level you mean… also the literature cited does not test and present clear result on circadian rhythm of cortisol and testosterone… please cite specific literature (e.g. for cortisol Doman et al. 1986).

Line 224: what concentration were the CV based on? Did you use high concentration and low concentration quality controls?

Line 238: Provide citation and version for R and Jamovi. Can you specify what you sued R for?

Results

Since the result section will be modified following the changes in statistical approach, I will not comment in details on this section. I would however recommend including a table of results to visualize which variables were included in the analyses, in particular in the final model. As presented it is hard to know if the approach is valid or not.

Discussion

Comments on the discussion will be provided on a revised version of the manuscript which revises the statistical approach.

Reviewer #2: The current study investigates whether inducing social stress and interacting with an opposite-sex individual affects participants hormone levels (testosterone and cortisol). It further investigates whether partnered and single participants respond differently to the induced stress or social situation.

My general impression is very positive. The manuscript is expertly written, the research questions are interesting, and the methods are appropriate. I really liked the limitations part in the discussion section (great that you discuss potential issues with salivary immunoassays).

This being said, I would like to share some thoughts that came to my mind while reading through the manuscript. The points raised do only address the results section and the rather complex analyses (that are sometimes hard to interpret). I really hope that my feedback is helpful and helps to improve the manuscript.

First, I am not sure whether I got this right, was the main analysis a 4-way interaction? Or did the authors compute several two-way interactions? This was not 100% clear to me, please specify what you did here in detail. For now, I assume an anova with a 4-way interaction (treatment group x sex x relationship status x saliva sample). If this is correct, I do have strong concerns regarding test power. Besides the fact that such higher order interactions are really complex and difficult to understand, three-way interactions require a much higher test power than two-way interactions (the problem is even more evident for four-way interactions). I would assume that these analyses are heavily underpowered and rather uninformative. Importantly, underpowered studies may not have the power to detect an expected (small) effect size, but, at the same time, underpowered studies that report significant effects might overestimate effect sizes which may lead to difficulties to replicate, due to a higher influence of random measurement error (Button et al., 2013).

Here is my suggestion how to deal with this issue:

I would recommend splitting the analyses, only investigating two-way interaction effects. Focus on the separate research questions you want to answer. To me, it seems as if the two-way interaction between treatment group and saliva sample is more of a method validation (does the TSST lead to higher cortisol/ testosterone levels?) and might be the first step. Then think about the second step and whether it is relevant to e.g. include a treatment variable in the analyses of the next research question. There were planned exploratory analyses regarding singles vs. partnered participants. I would recommend to do them separately from the main analyses, as a second step. Ask yourself if, for answering this specific research question, is it relevant to distinguish between participants who were in the TSST vs. control condition here? Is it relevant to distinguish between male and female participants here, or could this be a next step?

Second, please report (standardized) effect sizes and interpret your results accordingly. Reporting standardized effect sizes would make it easy for the reader to compare effect sizes across different analyses here and across different studies in general (and might also be helpful for future meta-analyses). It would also make it possible for the authors and for each reader to think about the reported effects in terms of effect sizes, not only based on p-values.

Minor:

- Page 17, line 252, something went wrong here (there are two p-values reported): “for control participants was not confirmed (t=1.24, p=1.0, p=0.002; Fig 1)”

- I do understand that you are not allowed to openly share your data, but I want to encourage you to think about sharing your analysis scripts (e.g. R script or SPSS syntax) and maybe, if possible, some material used in this study. This would make it easier for interested scholars to replicate this study in the future or to evaluate the analyses. Every little step in the direction of open and reproducible science helps.

Reference:

Button, K. S., Ioannidis, J. P., Mokrysz, C., Nosek, B. A., Flint, J., Robinson, E. S., & Munafò, M. R. (2013). Power failure: why small sample size undermines the reliability of neuroscience. Nature Reviews Neuroscience, 14, 365-376.

6. PLOS authors have the option to publish the peer review history of their article (what does this mean?). If published, this will include your full peer review and any attached files.

Reviewer #1: **Yes: **Cedric Girard-Buttoz

Reviewer #2: No

---

## [Author Response · Author response to Decision Letter 0]

14 Sep 2022

Reference: PONE-D-21-20363

Title: Hormonal responses to brief social interactions: The role of psychosocial stress and relationship status

Journal: PLoS ONE

Dear Dr. Sokolov,

We would like to begin by thanking you and the reviewers for your time and thoughtful comments on our paper, which have contributed to an improved manuscript. We paid very close attention to the comments of the reviewers while preparing this revision and have documented the changes we have made below. In order to assist you in reviewing our manuscript’s revisions, we have included each of the reviewer’s comments in the decision letter below, and our response to a comment is italicized and underlined. We have also edited the manuscript to meet the journal requirements mentioned in the decision letter, including formatting for style requirements, editing the use of “male” and “female” as nouns, and clarifying data availability in our updated cover letter.

Thank you in advance for your review!

Sincerely,

Nora (Nickels) McLean

Reviewer #1:

1. The introduction is relatively lengthy, not necessarily because it is too long but because it presents a suit of detailed findings from other studies with no clear direction of where the authors go. The main theories regarding hormonal responsiveness to social interactions are not named and the main theories published are not cited. The authors should revise the introduction to expand on theory and merge the exposition of previous findings into less lengthy phrasing.

In the introduction the interplay between Testosterone and cortisol in their hormonal regulation and the negative feedback effects that cortisol has on testosterone should be presented and discussed and incorporated in the hypothesis.

The authors have edited the introduction to expand on theory relevant to the research questions (e.g., the challenge hypothesis and the dual hormone hypothesis), as well as to help connect the expected fluctuations of these sex hormones to the research questions of interest. When possible, we have merged the exposition of previous findings to lead to less length phrasing. These recommendations have (very helpfully) led to a clearer flow between past findings and the current research questions.

2. Line 47: please nominatively introduce the challenge hypothesis (Wingfield et al. 1990) and expand citation to other studies which directly cited it in non-human primates or other animals (e.g. Muller & Wrangham 2004; Hirschenhauser & Oliveira 2006; Girard-Buttoz et al. 2015). 

The introduction of testosterone in the manuscript has been edited to incorporate the conceptual framework of the challenge hypothesis and supporting nonhuman evidence. 

3. One key aspect of this hypothesis is differences in testosterone reactivity to social interactions depending on paternity status. It appears therefore crucial to control for this parameter in the analysis beyond simply assessing if individuals are single or not.

In response to controlling for paternity and maternity status of men and women: only three participants reported having children at the time of the study; cortisol and testosterone levels did not significantly differ between those participants who had children and those who did not. 

4. Line 75: Here the authors point out the important effect of anticipation. However this parameter was not tested in their experiment since the participants did not know that they would be placed in a (albeit neutral) courtship opportunity situation. Such anticipatory principle is therefore not relevant to the current study.

This is a helpful comment; participants were not aware of the social interaction that is going to happen. We have removed the anticipatory principle from this paragraph. A social interaction itself can involve stress during the experience, particularly in a potentially social evaluative scenario; therefore, the rest of this paragraph’s introduction to a stressful component to courtship remains in the manuscript.

5. Line 93: Define ‘confederates’.

The term “confederates” has been removed from description of this particular citation, and instead replaced with female “individuals”. 

6. Line 121: present what cortisol is and highlight that it regulates the hormonal stress response in mammals including humans. In general, beyond providing a list of what testosterone and cortisol do and have been found to do it would be useful to provide clear cut presentation of what they regulate (e.g. muscle mass, aggressiveness for testosterone and energy mobilization for cortisol) and what they are expected to be sensitive too and why.

When introducing testosterone and cortisol, the authors have edited the manuscript to more thoroughly clarify the regulating functions of these hormones. This includes the clarification of important theories related to the two hormones, and better clarifying how their sensitivity may relate to our main and exploratory research questions. 

7. Line 123: What is presented in the following lines are not hypothesis but prediction and testable patterns… please formulate hypothesis first.

The final paragraph of the introduction section has been rewritten to clarify hypotheses and predictions. Specifically, it refers to the main hypotheses and exploratory analyses that are explored in separate sections of the rewritten results section of the revised manuscript. 

8. The methods are overall clear and well detailed. As previously, mention the authors should use GLMM rather than anova to clarify methodology and include more control variables. The authors should also run first a full model with all the interactions considered and compared it to a null model using an ANOVA and then reduce this full model removing non-significant interactions to obtain a final reduced model where the interactions and single effects are interpretable. In fact, single effects in interactions cannot be interpreted on their own. As previously mentioned, the authors should control for some important factors such as the paternity status of the men and the maternity status of the woman as well as whether they failed or passed the mathematical test which may have played a role in their hormonal stress response (one can imagine that failing is more stressful and that individual who failed may have more struggle solving mathematical problems hence be exposed to a higher stress).

Reviewers recommended two different statistical approaches for revisions for this manuscript. Based on concerns over power, the authors chose to split analyses and only investigate two-way interactions effects, focusing on specific research questions. Both the main research question and exploratory research questions were limited to two-way interaction effects to avoid concerns regarding test power, as recommended by Reviewer #2. The results section of the manuscript has been completely rewritten based on the new analysis. 

Addressing paternity and maternity status of men and women: only three participants reported having children at the time of the study; cortisol and testosterone levels did not significantly differ between those participants who had children and those who did not.

Addressing performance on the mathematical test: the mathematical test that is completed as part of the Trier Social Stress Test is a serial subtraction task. No participant successfully completed the entire task by counting down to zero (with or without a mistake). Data was not collected on how many mistakes were made for each participant. 

9. Write men and woman rather than male and female, it should be clear right from the start that you test humans.

Male and female are no longer used as nouns; they are replaced as adjective phrases or with men and women in the manuscript.

10. Line 147: Clarify what is SES

SES has been clarified as socioeconomic status in the manuscript. 

11. Line 217: Hormonal level is too general please indicate which level you mean… also the literature cited does not test and present clear result on circadian rhythm of cortisol and testosterone… please cite specific literature (e.g. for cortisol Doman et al. 1986).

Cortisol and testosterone levels have been specified, and additional specific literature has been cited regarding diurnal rhythms and methodological procedures. 

12. Line 224: what concentration were the CV based on? Did you use high concentration and low concentration quality controls?

Assay description in the method section of the manuscript has been edited to include specifics about both intra- and inter-assay CV calculations. 

13. Line 238: Provide citation and version for R and Jamovi. Can you specify what you sued R for?

For the updated analyses, only jamovi was used. The citation and version have been added to the manuscript. 

14. Since the result section will be modified following the changes in statistical approach, I will not comment in details on this section. I would however recommend including a table of results to visualize which variables were included in the analyses, in particular in the final model. As presented it is hard to know if the approach is valid or not.

Reviewers recommended two different statistical approaches for revisions for this manuscript. Based on concerns over power, the authors chose to split analyses and only investigate two-way interactions effects, focusing on specific research questions. Both main research questions and exploratory research questions were limited to two-way interaction effects to avoid concerns regarding test power, as recommended by Reviewer #2. The results section of the manuscript has been completely rewritten based on the new analysis.

Reviewer #2:

My general impression is very positive. The manuscript is expertly written, the research questions are interesting, and the methods are appropriate. I really liked the limitations part in the discussion section (great that you discuss potential issues with salivary immunoassays).

This being said, I would like to share some thoughts that came to my mind while reading through the manuscript. The points raised do only address the results section and the rather complex analyses (that are sometimes hard to interpret). I really hope that my feedback is helpful and helps to improve the manuscript.

1. First, I am not sure whether I got this right, was the main analysis a 4-way interaction? Or did the authors compute several two-way interactions? This was not 100% clear to me, please specify what you did here in detail. For now, I assume an anova with a 4-way interaction (treatment group x sex x relationship status x saliva sample). If this is correct, I do have strong concerns regarding test power. Besides the fact that such higher order interactions are really complex and difficult to understand, three-way interactions require a much higher test power than two-way interactions (the problem is even more evident for four-way interactions). I would assume that these analyses are heavily underpowered and rather uninformative. Importantly, underpowered studies may not have the power to detect an expected (small) effect size, but, at the same time, underpowered studies that report significant effects might overestimate effect sizes which may lead to difficulties to replicate, due to a higher influence of random measurement error (Button et al., 2013). Here is my suggestion how to deal with this issue:

I would recommend splitting the analyses, only investigating two-way interaction effects. Focus on the separate research questions you want to answer. To me, it seems as if the two-way interaction between treatment group and saliva sample is more of a method validation (does the TSST lead to higher cortisol/ testosterone levels?) and might be the first step. Then think about the second step and whether it is relevant to e.g. include a treatment variable in the analyses of the next research question. There were planned exploratory analyses regarding singles vs. partnered participants. I would recommend to do them separately from the main analyses, as a second step. Ask yourself if, for answering this specific research question, is it relevant to distinguish between participants who were in the TSST vs. control condition here? Is it relevant to distinguish between male and female participants here, or could this be a next step?

Reviewers recommended two different statistical approaches for revisions for this manuscript. Based on concerns over power, the authors chose to split analyses and only investigate two-way interactions effects, focusing on specific research questions. Both main research questions and exploratory research questions were limited to two-way interaction effects to avoid concerns regarding test power, as recommended by Reviewer #2. The results section of the manuscript has been completely rewritten based on the new analysis.

2. Second, please report (standardized) effect sizes and interpret your results accordingly. Reporting standardized effect sizes would make it easy for the reader to compare effect sizes across different analyses here and across different studies in general (and might also be helpful for future meta-analyses). It would also make it possible for the authors and for each reader to think about the reported effects in terms of effect sizes, not only based on p-values.

Standardized effect sizes and interpretations are now reported for all significant main and interaction effects reported in the manuscript. 

3. Page 17, line 252, something went wrong here (there are two p-values reported): “for control participants was not confirmed (t=1.24, p=1.0, p=0.002; Fig 1)”

This error has been removed, due to new analyses replacing the prior results section of the original manuscript. 

4. I do understand that you are not allowed to openly share your data, but I want to encourage you to think about sharing your analysis scripts (e.g. R script or SPSS syntax) and maybe, if possible, some material used in this study. This would make it easier for interested scholars to replicate this study in the future or to evaluate the analyses. Every little step in the direction of open and reproducible science helps.

.omt files (jamovi script files, written in R script) are being submitted with the revision. Separate files for main and exploratory analyses are prepared and have been submitted with the revision as a supporting information zip file.

---

## [Decision Letter · Decision Letter 1]

29 Nov 2022

PONE-D-21-20363R1Hormonal responses to brief social interactions: The role of psychosocial stress and relationship statusPLOS ONE

Dear Dr. Nickels,

thank you for submitting your revised manuscript to PLOS ONE.  After careful consideration, we feel that it has merit but as it currently stands, still has to be revised in order to fully meet PLOS ONE’s publication criteria.  Therefore, we invite you to submit a revised version of the manuscript that addresses the points raised by Reviewer 1 listed below, especially concerning the technical aspects of your work.

Please make sure to carefully, in the point-by-point manner respond to the issues raised and run in-depth check of the main text of your submission.

Please submit your revised manuscript within six months from this date as thereafter, any revision has to be considered a new submission. If you will need more time than this to complete your revisions, please reply to this message or contact the journal office at plosone@plos.org. Please include the following items when submitting your revised manuscript:A rebuttal letter that responds to each point raised by the academic editor and reviewer(s). You should upload this letter as a separate file labeled 'Response to Reviewers'.A marked-up copy of your manuscript that highlights changes made to the original version. You should upload this as a separate file labeled 'Revised Manuscript with Track Changes'.An unmarked version of your revised paper without tracked changes. You should upload this as a separate file labeled 'Manuscript'.If applicable, we recommend that you deposit your laboratory protocols in protocols.io to enhance the reproducibility of your results. Protocols.io assigns your protocol its own identifier (DOI) so that it can be cited independently in the future. For instructions see: https://journals.plos.org/plosone/s/submission-guidelines#loc-laboratory-protocols. Additionally, PLOS ONE offers an option for publishing peer-reviewed Lab Protocol articles, which describe protocols hosted on protocols.io. Read more information on sharing protocols at https://plos.org/protocols?utm_medium=editorial-email&utm_source=authorletters&utm_campaign=protocols.

We look forward to receiving your revised manuscript.  Thank you for choosing PLOS ONE for reporting your research.

Kind regards,

Sasha

Alexander N. 'Sasha' Sokolov, Ph.D.

Academic Editor

PLOS ONE

Journal Requirements:

Reviewers' comments:

Reviewer's Responses to Questions

**Comments to the Author**

1. If the authors have adequately addressed your comments raised in a previous round of review and you feel that this manuscript is now acceptable for publication, you may indicate that here to bypass the “Comments to the Author” section, enter your conflict of interest statement in the “Confidential to Editor” section, and submit your "Accept" recommendation.

Reviewer #1: (No Response)

Reviewer #2: All comments have been addressed

2. Is the manuscript technically sound, and do the data support the conclusions?

Reviewer #1: Yes

Reviewer #2: Yes

3. Has the statistical analysis been performed appropriately and rigorously? 

Reviewer #1: Yes

Reviewer #2: Yes

4. Have the authors made all data underlying the findings in their manuscript fully available?

Reviewer #1: Yes

Reviewer #2: No

5. Is the manuscript presented in an intelligible fashion and written in standard English?

Reviewer #1: Yes

Reviewer #2: Yes

6. Review Comments to the Author

Reviewer #1: In this revised version, the authors have made a great work at reformatting the introduction and have now a nicely introduced paper. The paper has improved overall and is now in a nice shape. I have still a few comments and the authors should conduct an in depth detailed check of typos and bracket errors since I have seen a few and did not point each of them in detail. I am still advocating for different statistical approaches but what the author used, albeit outdated and not in line with current analytical standard, is not incorrect. The method lacks a few details regarding the hormonal measurements and the choice of the time frame for sample collection. The term confederates should either be explained clearly or removed throughout.

Introduction:

Line 105: “consistent” not “consisten”.

Methods

Line 191: by ‘confederates’ do you mean ‘judges’? I thought you removed the term confederate throughout the manuscript, maybe this is a typo or has been forgotten here.

Line 210: same here and throughout method description, change confederate to another term or define it somewhere.

Line 230-232: Can you clarify this sentence?

Line 236: here or before when describing the timing of saliva sample can you elaborate on the clearance delay of both cortisol and testosterone in saliva and give background citation to support the adequacy of your protocol?

Line 246: what does ‘base on concentration’ mean here. Explain here already that you used high and low QCs? Give CVs for each separately. Also why dii you calculate intra-assay CV as “an average value calculated from the individual CVs from all the plate duplicates”. You should use the average CV only from the duplicates of the QCs and not from your sample as you do not know if theyare contaminated or there were sampling error. Obviously the ducplicates with high CV should be removed and reanalyzed. Can you clarify if you did that and what was your cuttof for reanalysis?

Line 263: Please specify the distribution of single and ‘in relationship’ individuals across the TSST and the control conditions. #

Results

Line 273-276: This is methods and not results. As previously mention the authors should conduct GLMM analyses, ANOVA are highly outdated. With GLMM individual could be entered as random effect to control for the repeated sampling on each individual. I will not reiterate in details my recommendations from the first review but the authors should provide a test that their full model is significantly different from a null model comprising only the control predictors and not the test predictors.

Line 358: close the citation bracket

Discussion

The discussion as it stands is interesting and acknowledges nicely the limitations of the study but remains relatively shallow. The discussion should loop back to the theories introduced in the introduction (challenge hypothesis and dual hormone hypothesis) and specifically state whether the results here corroborate (even if only partly) or not each hypothesis.

Line 292: did you ask the participants specifically how they felt the interaction went and if they had the impression to flirt?

Reviewer #2: I think the authors did a good job and I am convinced by their detailed revisions. Indeed, I feel like they have taken almost all of my and the other reviewer’s comments into account or have at least explained their decisions in a reasonable manner. I appreciate their newly reported analyses and the work they put into rewriting their entire results part.

7. PLOS authors have the option to publish the peer review history of their article (what does this mean?). If published, this will include your full peer review and any attached files.

Reviewer #1: **Yes: **Cédric Girard-Buttoz

Reviewer #2: No

---

## [Author Response · Author response to Decision Letter 1]

14 Jan 2023

The response to specific reviewer and editor comments can be found in the response to reviewers document submitted in this revision. Thank you!

---

## [Decision Letter · Decision Letter 2]

1 Jun 2023

Hormonal responses to brief social interactions: The role of psychosocial stress and relationship status

PONE-D-21-20363R2

Dear Dr. McLean,

thank you for your patience with processing your revised manuscript.  We are pleased to inform you that the revision has been judged suitable for publication and will be formally accepted for publication once it meets all outstanding technical requirements. 

As a minor comment please consider to re-formulate your Data Availability statement, exchanging the final phrase on possible data provision to qualified researchers upon request to the Social and Behavioral Sciences Institutional Review Board Ethics Committee of your University with the explanation why the data cannot be made available publicly. 

Thank you again for considering PLOS ONE for reporting your work.

Kind regards,

Sasha

Alexander N. 'Sasha' Sokolov, Ph.D.

Academic Editor

PLOS ONE

Additional Editor Comments (optional):

Reviewers' comments:

Reviewer's Responses to Questions

**Comments to the Author**

1. If the authors have adequately addressed your comments raised in a previous round of review and you feel that this manuscript is now acceptable for publication, you may indicate that here to bypass the “Comments to the Author” section, enter your conflict of interest statement in the “Confidential to Editor” section, and submit your "Accept" recommendation.

Reviewer #1: All comments have been addressed

Reviewer #2: All comments have been addressed

2. Is the manuscript technically sound, and do the data support the conclusions?

Reviewer #1: Yes

Reviewer #2: Yes

3. Has the statistical analysis been performed appropriately and rigorously? 

Reviewer #1: Yes

Reviewer #2: Yes

4. Have the authors made all data underlying the findings in their manuscript fully available?

Reviewer #1: No

Reviewer #2: No

5. Is the manuscript presented in an intelligible fashion and written in standard English?

Reviewer #1: Yes

Reviewer #2: Yes

6. Review Comments to the Author

Reviewer #1: (No Response)

Reviewer #2: I didn't have any more requests in the previous round and I can only reiterate that I am satisfied with the author's revisions. In my opinion, this is an interesting manuscript that is worth publishing.

7. PLOS authors have the option to publish the peer review history of their article (what does this mean?). If published, this will include your full peer review and any attached files.

Reviewer #1: No

Reviewer #2: No

---

## [Editor Report · Acceptance letter]

15 Jun 2023

PONE-D-21-20363R2 

Hormonal responses to brief social interactions: The role of psychosocial stress and relationship status 

Dear Dr. Nickels McLean:

I'm pleased to inform you that your manuscript has been deemed suitable for publication in PLOS ONE. Congratulations! Your manuscript is now with our production department. 

Kind regards, 

on behalf of

Dr. Alexander N. Sokolov 

Academic Editor

PLOS ONE